EMBO
Molecular Medicine

# A single epidermal stem cell strategy for safe *ex vivo* gene therapy

Stéphanie Droz-Georget Lathion[1,2], Ariane Rochat[1,2], Graham Knott[3], Alessandra Recchia[4], Danielle Martinet[5], Sara Benmohammed[6], Nicolas Grasset[1,2], Andrea Zaffalon[1,2], Nathalie Besuchet Schmutz[5], Emmanuelle Savioz-Dayer[1,2], Jacques Samuel Beckmann[5,6], Jacques Rougemont[7], Fulvio Mavilio[4,8] & Yann Barrandon[1,2,*]

## Abstract

There is a widespread agreement from patient and professional organisations alike that the safety of stem cell therapeutics is of paramount importance, particularly for *ex vivo* autologous gene therapy. Yet current technology makes it difficult to thoroughly evaluate the behaviour of genetically corrected stem cells before they are transplanted. To address this, we have developed a strategy that permits transplantation of a clonal population of genetically corrected autologous stem cells that meet stringent selection criteria and the principle of precaution. As a proof of concept, we have stably transduced epidermal stem cells (holoclones) obtained from a patient suffering from recessive dystrophic epidermolysis bullosa. Holoclones were infected with self-inactivating retroviruses bearing a *COL7A1* cDNA and cloned before the progeny of individual stem cells were characterised using a number of criteria. Clonal analysis revealed a great deal of heterogeneity among transduced stem cells in their capacity to produce functional type VII collagen (COLVII). Selected transduced stem cells transplanted onto immunodeficient mice regenerated a non-blistering epidermis for months and produced a functional COLVII. Safety was assessed by determining the sites of proviral integration, rearrangements and hit genes and by whole-genome sequencing. The progeny of the selected stem cells also had a diploid karyotype, was not tumorigenic and did not disseminate after long-term transplantation onto immunodeficient mice. In conclusion, a clonal strategy is a powerful and efficient means of by-passing the heterogeneity of a transduced stem cell population. It guarantees a safe and homogenous medicinal product, fulfilling the principle of precaution and the requirements of regulatory affairs. Furthermore, a clonal strategy makes it possible to envision exciting gene-editing technologies like zinc finger nucleases, TALENs and homologous recombination for next-generation gene therapy.

**Keywords** cell therapy; regulatory affairs; stem cells; wound healing
**Subject Categories** Regenerative Medicine; Stem Cells; Skin

See also: **JC Larsimont & C Blanpain** (April 2015)

## Introduction

*Ex vivo* gene therapy can permanently cure debilitating hereditary diseases (Hacein-Bey-Abina *et al*, 2002; Mavilio *et al*, 2006; Ott *et al*, 2006; Gargioli *et al*, 2008; Naldini, 2009; Mavilio, 2010; Tedesco *et al*, 2011; Aiuti *et al*, 2013; Biffi *et al*, 2013). Therapeutical success has been obtained in pioneer trials using genetically corrected human bone marrow stem cells to treat patients suffering from X-linked severe combined immunodeficiency (SCID) (Hacein-Bey-Abina *et al*, 2002), X-linked adrenoleukodystrophy (ALD) (Cartier *et al*, 2009) and SCID-adenosine deaminase (ADA-SCID) (Aiuti *et al*, 2009). However, unexpected complications like T-cell leukaemia have raised concerns (Hacein-Bey-Abina *et al*, 2003; Howe *et al*, 2008) about the safety of *ex vivo* gene therapy (Williams & Baum, 2003). Complications result from insertional mutagenesis together with clonal dominance (Hacein-Bey-Abina *et al*, 2008; Howe *et al*, 2008). Hence, the population of recombinant stem cells should be characterised before it is transplanted (Halme & Kessler, 2006; Fink, 2009). However, most tissue stem cells (e.g. hematopoietic and neural stem cells) cannot be efficiently expanded in culture by

1 Department of Experimental Surgery, Lausanne University Hospital (CHUV), Lausanne, Switzerland
2 Laboratory of Stem Cell Dynamics, Ecole Polytechnique Fédérale de Lausanne (EPFL), Lausanne, Switzerland
3 Interdisciplinary Center for Electron Microscopy, Faculty of Life Sciences EPFL, Lausanne, Switzerland
4 Department of Life Sciences, University of Modena and Reggio Emilia, Modena, Italy
5 Service de Génétique Médicale, Lausanne University Hospital (CHUV), Lausanne, Switzerland
6 Department of Medical Genetics, Université de Lausanne, Lausanne, Switzerland[†]
7 Bioinformatics and Biostatistics Core Facility, Faculty of Life Sciences EPFL, Lausanne, Switzerland[†]
8 Genethon, Evry, France
*Corresponding author. Tel: +41 21 314 24 61; Fax: +41 21 314 24 68; E-mail: yann.barrandon@epfl.ch
[†]Correction added on 10 March 2015, after first online publication: author affiliations have been corrected.

 

present technologies. To compensate for this limitation, integration sites have been documented in engrafted cells but it is only informative *a posteriori* (Aiuti *et al*, 2013).

Human epidermal stem cells are privileged among adult (tissue) stem cells because they can be efficiently expanded *ex vivo* (Gallico *et al*, 1984; Rochat *et al*, 2013). The technology is based on the use of a lethally irradiated feeder layer of mouse 3T3-J2 cells that provides the necessary microenvironment to promote stem cell expansion (Rheinwald & Green, 1975; Barrandon & Green, 1987; Barrandon *et al*, 2012). Using this technology, it is possible to isolate enough epidermal stem cells from a small skin biopsy to generate large amounts of keratinocytes when cells are properly cultured; our laboratory is routinely using a strain of human diploid keratinocytes (YF29) isolated more than 25 years ago from the foreskin of a newborn. Most importantly, epidermal stem cells are used worldwide to treat extensive third-degree burn wounds to permanently restore epidermis (Gallico *et al*, 1984; Pellegrini *et al*, 1999; Ronfard *et al*, 2000; Chua *et al*, 2008; Cirodde *et al*, 2011). The use of autologous cultured epithelium is approved by FDA (Food and Drug Administration) as a humanitarian use device and is commercially available worldwide. Moreover, it is possible to efficiently clone epidermal stem cells and to obtain a large progeny from a single stem cell, a property that we have used to characterise growth capabilities and transplantability (Barrandon & Green, 1987; Rochat *et al*, 1994; Mathor *et al*, 1996; Claudinot *et al*, 2005; Majo *et al*, 2008; Bonfanti *et al*, 2010). Furthermore, human epidermal stem cells can be efficiently transduced by means of recombinant retroviruses to produce proteins of medical interest (Morgan *et al*, 1987; Mathor *et al*, 1996; Warrick *et al*, 2012). This was used to transplant a patient suffering from junctional epidermolysis bullosa with an engineered recombinant cultured epidermis producing laminin 5 (Mavilio *et al*, 2006; De Rosa *et al*, 2014). Taken together, these observations lead us to consider the feasibility of a single stem cell strategy for *ex vivo* gene therapy of debilitating hereditary skin disease while assessing its medical safety before clinical use.

To demonstrate the feasibility of our strategy, we have selected severe generalised recessive dystrophic epidermolysis bullosa (Hallopeau-Siemens RDEB, OMIM 226600) as a model system for the following reasons. First, RDEB is a genodermatosis for which there is no curative treatment. RDEB is characterised by an extremely severe blistering due to poor adherence of epidermis to the dermis caused by deficient type VII collagen (COLVII), the major component of the anchoring fibrils (Bruckner-Tuderman *et al*, 1999; Fine *et al*, 2008). As a consequence, RDEB patients have extensive chronic wounds that can ultimately lead to death by invasive squamous cell carcinomas (Fine *et al*, 2009). Second, the severity of the disease has led to major therapeutic adventures like the transplantation of allogeneic bone marrow stem cells (Wagner *et al*, 2010), resulting in several patients' death (Tolar & Wagner, 2012, 2013) and other unconventional therapeutic alternatives (Woodley *et al*, 2007; Wong *et al*, 2008; Remington *et al*, 2009; Siprashvili *et al*, 2010; Itoh *et al*, 2011; Tolar *et al*, 2011). Third, we have demonstrated that a clone of human keratinocytes can produce COLVII and participate in the formation of anchoring fibrils (Regauer *et al*, 1990). Fourth, we have access to patients with well-characterised mutations, and among them a patient with a homozygous insertion–deletion resulting in a premature stop codon and absence of functional COLVII (Hilal *et al*, 1993; Hovnanian *et al*, 1997).

Our strategy is inspired by the protocols and guidelines developed by the biotechnology industry and regulatory affairs to produce medicinal proteins by means of genetically engineered mammalian cells [http://www.ich.org/fileadmin/Public_Web_Site/ICH_Products/Guidelines/Quality/Q5D/Step4/Q5D_Guideline.pdf (ICH, 1997); http://www.ich.org/fileadmin/Public_Web_Site/ICH_Products/Guidelines/Quality/Q7/Step4/Q7_Guideline.pdf (ICH, 2000);

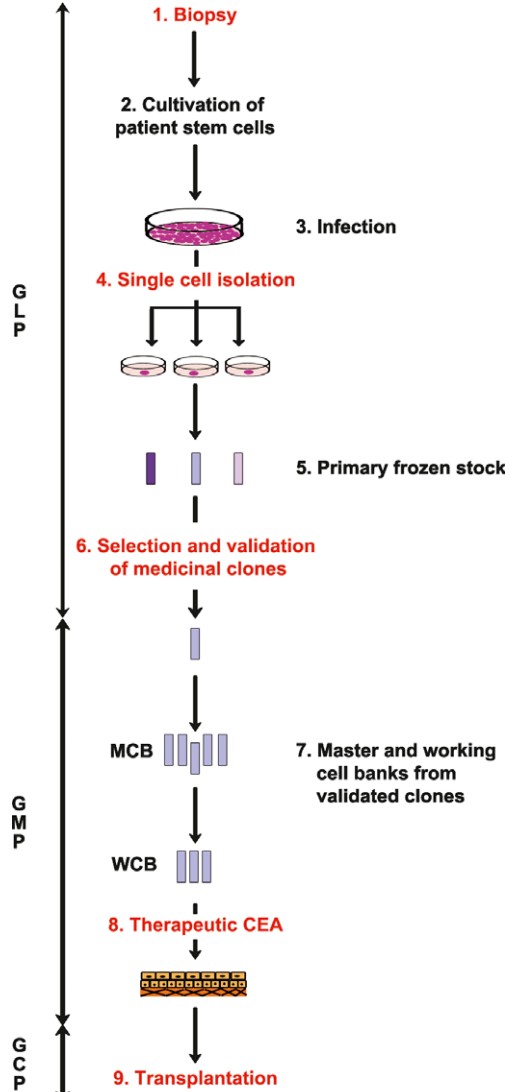

**Figure 1.  Strategy to perform *ex vivo* gene therapy from a single epidermal stem cell.**
Schematic strategy to produce a performant and safe gene therapy product from a single autologous epidermal stem cell. (1) A biopsy is obtained from the patient to isolate epidermal stem cells that are then expanded under appropriate conditions (2). An aliquot of the culture is infected with the *ad hoc* recombinant shuttle virus (3). Single cells are then isolated (4) and clones expanded to built a frozen stem cell bank (5). In parallel, an aliquot of each clone is expanded to select for a clone fulfilling the criteria described in Table 1. After validation (6), the approved clone is thawed and expanded to create master and working cell banks in a GMP facility (7). Genetically modified CEA are then produced (8) and grafts are transplanted onto the patient (9). MCB, master cell bank; WCB, working cell bank; CEA, cultured epidermal autografts; GLP, good laboratory practice; GMP, good manufacturing practice; GCP, good clinical practice.

http://www.isscr.org/docs/guidelines/isscrglclinicaltrans.pdf (ISSCR, 2008)]. The best clone following GLP (good laboratory practices) is first fully characterised and then transferred to GMP (good manufacturing practices) to prepare the master and working cell banks. The strategy for *ex vivo* gene therapy (Fig 1) is firstly isolation of epidermal stem cells from a patient's biopsy (step 1) and cultivation (step 2) before being permanently transduced by means of disease-specific viral shuttle vectors (step 3). Single cells are then isolated to obtain clones (step 4) that are expanded before they are individually frozen (step 5). In parallel, a small aliquot of each clone is expanded for further characterisation and validation (step 6). Once a clone fulfils the strict functionality and safety requirements described in Table 1, master and working cell banks are prepared in a GMP facility (step 7) in which genetically corrected autologous cultured epithelia (CEA) are also produced (step 8). These CEA are then transferred to the clinic and transplanted onto the patient (step 9). Our experiments have demonstrated that it is possible to produce enough genetically corrected autologous transplants from a single human epidermal stem cell for a pilot clinical trial fulfilling strict safety criteria.

# Results

### Identification of epidermal stem cells in the skin of an RDEB patient

Recessive dystrophic epidermolysis bullosa keratinocytes were isolated from a small skin biopsy obtained from a 4-year-old patient with a homozygous insertion–deletion in the type VII collagen gene (*COL7A1*) leading to a premature stop codon in the fibronectin 5 domain and to the formation of severely truncated type VII collagen (Hilal *et al*, 1993). RDEB clonogenic keratinocytes were cultivated onto lethally irradiated 3T3-J2 cells according to standard procedures

used for cell therapy of third-degree burn wounds (Pellegrini *et al*, 1999; Ronfard *et al*, 2000). No COLVII was detected in skin biopsies, nor in cultured fibroblasts and keratinocytes (Supplementary Fig S1). Because RDEB keratinocytes have been described as poor growers in the literature (Morley *et al*, 2003), we first determined the lifespan of RDEB clonogenic keratinocytes obtained from an early passage. Cells were subcultured once a week for 5 months (Fig 2A), and the percentage of growing colonies determined as it is the most reliable indicator of the growth capacity of a keratinocyte culture (Rochat *et al*, 2013). As expected, colony-forming efficiency (CFE) and the number of growing colonies decreased with time but at a similar rate to that of healthy keratinocytes, demonstrating that the growth potential of RDEB cells was not different from that of healthy cells of similar age. To further analyse the growth potential of RDEB clonogenic keratinocytes, we cloned individual RDEB keratinocytes that had already undergone five subcultures and at least 30 doublings (patient cells divided on average once a day) (Barrandon & Green, 1987). This experiment demonstrated the presence of holoclones (6% of clones) (Fig 2B), considered as the phenotype of human keratinocyte stem cells in culture (Barrandon & Green, 1987; Mathor *et al*, 1996; Rama *et al*, 2010; Rochat *et al*, 2013). Our data demonstrate that RDEB keratinocyte stem cells can be expanded in culture and cloned as are healthy keratinocytes.

### Genetic correction of RDEB epidermal stem cells

RDEB keratinocytes were infected as a passage IV mass culture with a suspension of self-inactivating (SIN) retroviruses bearing a *COL7A1* cDNA under the control of a minimal human elongation factor 1α (*EF1α*) promoter (Supplementary Fig S2) as previously described (Titeux *et al*, 2010). We first demonstrated that the infection procedure was compatible with stem cell maintenance (Supplementary Fig S3) and that on average thirty-five to forty-two per cent of the infected RDEB keratinocytes were positive for COLVII by immunostaining (Supplementary Fig S4). Next, we manually

**Table 1.  Selection criteria for safety assessments of medicinal epidermal stem cells.**

|  | Selection criteria | Assays | Levels of confidence | |
|---|---|---|---|---|
|  |  |  | Mass culture | Clonal culture |
| Quality of medicinal product | High growth potential | Clonal analysis | Low | High |
|  | Production of the protein of interest | Western blotting/Immunocytochemistry | Low | High |
|  | Long-term tissue regeneration | *In vivo* transplantation onto immunodeficient mice | Low | High |
|  | Long-term correction of the disease | *In vivo* transplantation onto immunodeficient mice | Low | High |
| Safety of medicinal product | No immortalisation | Serial passaging (cellular lifespan) | Low | High |
|  |  | Western blotting (G1 checkpoint) | Low | High |
|  |  | Karyotyping | Low | High |
|  | No tumorigenic potential | Subcutaneous injection into athymic mice | Low | High |
|  | Determination of proviral integrations | Ligation-mediated PCR | Low | Medium |
|  |  | Fluorescence *in situ* hybridisation | Low | High |
|  |  | Whole-genome sequencing | Low | High |
|  | No dissemination of genetically modified human stem cells | Organ analysis of transplanted immunodeficient mice | Low | High |

Selection criteria used to determine efficacy and safety of corrected stem cells before transplantation. These could be performed on mass culture or on single cell expansion. We determined the degree of reliability of each assay as low, medium and high. A clonal strategy gives a higher level of safety.

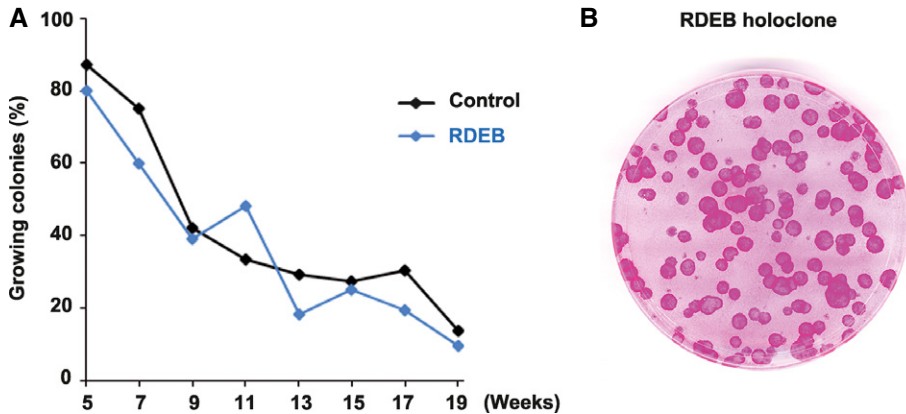

**Figure 2.   Extensive growth potential of recessive dystrophic epidermolysis bullosa (RDEB) epidermal keratinocytes.**

A   Keratinocytes were isolated from the skin of a 4-year-old patient with severe-generalised RDEB linked to homozygous insertion–deletion in *COL7A1* (Hilal *et al*, 1993). Cultured RDEB cells (blue line) were serially passaged for more than 4 months, displaying a growth potential similar to non-diseased control cells (YF29) isolated from the foreskin of a newborn (black line). To calculate the percentage of growing colonies, 100 to 1,000 cells were plated into indicator dishes at each passage. Cells were grown for 12 days, fixed and stained with rhodamine B. Colonies were scored as growing or aborted (Barrandon & Green, 1987).

B   Clonal analysis demonstrated the presence of stem cells (holoclones) in a passage VII RDEB culture (95% of growing colonies).

isolated one hundred and fifty single cells with a Pasteur pipette under an inverted microscope (Barrandon & Green, 1985). Sixty-seven clones were obtained, fifteen of which were obviously non-growing (paraclones). Each of the remaining fifty-two clones was transferred individually in several Petri dishes, first to expand the population, second to determine the clonal type (Barrandon & Green, 1987) and third to determine the production of COLVII. Three clones were classified as holoclones, forty-two as meroclones and three as paraclones; four clones were lost during cultivation for technical problems (Supplementary Table S1). COLVII was immuno-detected in two holoclones (out of three), eighteen meroclones (out of forty-two) and three paraclones (all positive) (Supplementary Table S1). This demonstrated that keratinocytes with extensive or restricted growth potential were equally transduced and that COLVII expression was independent of clonal type. Next, we thoroughly characterised COLVII-positive clones (cl.6, cl.17, cl.22, cl.58 and cl.61) and COLVII-negative clones (cl.3, cl.24 and cl.54) (Fig 3A). qPCR experiments confirmed that the expression of *COL7A1* was variable in different clones (Fig 3B), with levels of mRNAs varying from twofold to fiftyfold (clone 3 and clone 58, respectively) compared to uninfected RDEB keratinocytes. As expected, the life-span of the individual clones was different, holoclones having a higher growth potential than meroclones (Supplementary Fig S5). We then showed that transduced keratinocytes expressed COLVII until the last subculture, eleven weeks after the start of the experiment (Supplementary Fig S6). Transduced *COL7A1* cDNAs are known to frequently rearrange in contrast to other collagens (F. Mavilio, unpublished data); therefore, we performed Southern blots on genomic DNA obtained from several transduced clones using a *COL7A1*-specific probe (Fig 3C). The retroviral producer cloned line Flp293A-E1aColVII1 was used as a positive control. Bands corresponding to endogenous *COL7A1* (16 kb) and proviral DNA (9.6 kb) were observed in control cells, whereas bands corresponding to the expected proviral DNA and rearranged proviral DNA were observed in clones 6 and 54. These rearrangements were not clearly detected in the infected cell pools from which clones 6 and 54 were

isolated (lane 3); this does not mean that there was no rearrangement in the mass culture containing thousands of transduced stem cells, but rather that the use of a genetically homogenous population (clones) increased the threshold of detection. Next, the culture supernatants of transduced keratinocytes were analysed to determine whether COLVII was secreted. Only clone 6 correctly secreted COLVII while clone 54 did not (Fig 3D), further emphasising that a mass culture of transduced cells is vastly heterogeneous. This observation by itself justifies the clonal strategy.

### RDEB-corrected stem cells generate a functional self-renewing epidermis

The first selection criterion for suitable gene therapy is the high growth potential of the COLVII-producing cells (corrected stem cells). This is a *sine qua non* condition to obtain a sufficient number of recombinant grafts to treat a patient. We thus performed serial transfer analysis of corrected clone 6 and compared it to COLVII non-producing clone 54 (uncorrected) (Fig 4A). Clones 6 and 54 had an extended lifespan as expected from their clonal type. Clone 6 could undergo eleven serial transfers from the day of cloning, which equals to fifty-nine population doublings, yielding a theoretical progeny of up to $5.7 \times 10^{17}$ cells (Fig 4B). Collectively, these experiments demonstrated that a single transduced stem cell (holoclone) could generate a progeny large enough to produce medicinal CEA to treat wide areas of diseased skin.

Genetically corrected keratinocytes obtained from a single trans-duced stem cell were challenged in a long-term transplantation assay to determine whether they could regenerate a functional epidermis. Cells were plated onto a fibrin-based matrix containing autologous untransduced RDEB fibroblasts and were grown to confluence, and engineered epithelia were transplanted onto immunodeficient SCID mice (Larcher *et al*, 2007). Biopsies of the engrafted area were then performed at various time points and processed for histology to confirm the human origin of the epidermis (HLA-1

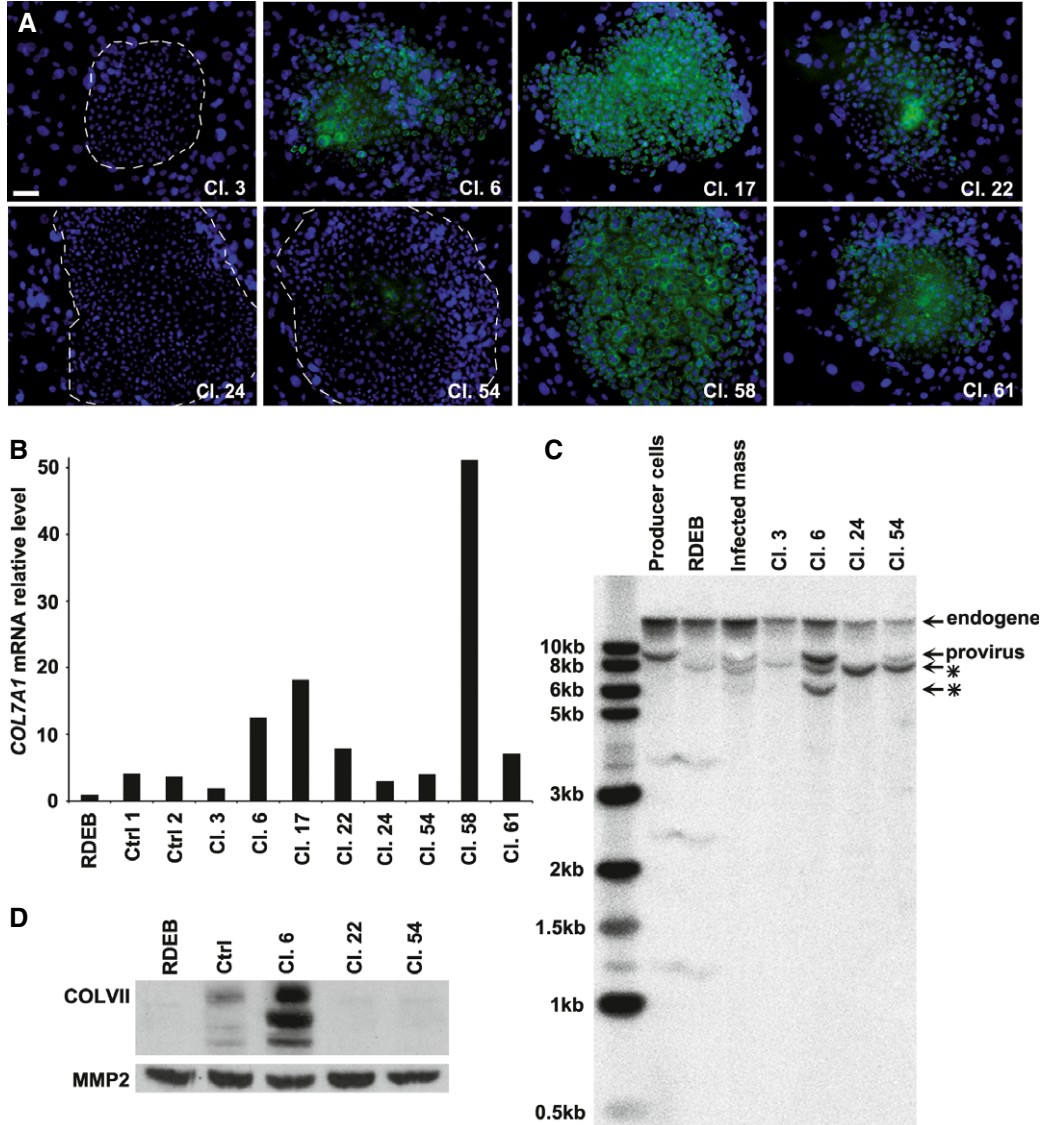

**Figure 3.  Isolation of genetically corrected recessive dystrophic epidermolysis bullosa (RDEB) epidermal stem cells.**

Single cells were isolated from a mass culture (passage V) of RDEB keratinocytes infected with SIN retroviruses bearing a *COL7A1* cDNA. Clonal types were determined (Barrandon & Green, 1987) and listed in Supplementary Table S1. Growing clones were expanded for further characterisation.

A   COLVII detection in clones by immunostaining. COLVII expression (green) was detectable in some clones (6, 17, 22, 58 and 61) and not in others (3, 24 and 54); nuclei were stained with Hoechst 33342 (blue). Dotted lines delimit the periphery of keratinocyte colonies from the surrounding irradiated 3T3-J2 feeder cells. Scale bar: 50 µm.

B   Quantitative RT–PCR analysis of *COL7A1* expression in transduced clones compared to untransduced RDEB keratinocytes. All clones shown in (A) were transduced but expressed different levels of *COL7A1* transcripts. Clones 6, 17, 22, 54, 58 and 61 expressed higher levels of *COL7A1* than control RDEB cells and keratinocytes obtained from healthy donors (YF29 and OR-CA, control 1 and 2, respectively). The level of *COL7A1* expression in the RDEB untransduced cells was referenced as 1.

C   Determination of proviral rearrangements in transduced clones. A Southern blot was performed using genomic DNA of RDEB cells, clones and the infected mass culture from which the clones were isolated. Genomic DNA was digested with EcoRV and SpeI that cut at the 3′ and 5′ end of the provirus (Supplementary Fig S2) and hybridised with a 907-bp *COL7A1* probe radiolabelled with $^{32}$P isotope. The upper band corresponded to the endogenous signal. The retroviral producer line Flp293A-E1aColVII1 was used as a control for the digested 9.6-kb provirus (proviral signal). Smaller bands corresponded to rearranged proviruses marked with an asterisk.

D   Identification of stem cells producing COLVII. Western blotting revealed that only clone 6 secreted COLVII in the culture supernatant, while clone 54 and surprisingly clone 22 did not (see A). RDEB cells were used as a negative control and healthy donor cells as a positive control. The secreted matrix metalloproteinase 2 (MMP2) was used as a loading control.

positive) and for immunodetection of COLVII (Fig 4). Untransduced RDEB keratinocytes generated a normal epidermis, which was blistering and poorly attached onto the underlying dermis, consistent with an absence of COLVII (Fig 4C). *COL7A1* transduced clone 22,

which did not secrete COLVII (Fig 3D), generated an epidermis that behaved like untransduced RDEB keratinocytes. On the other hand, corrected clone 6 formed a normal epidermis that adhered to the underlying dermis and did not blister for at least 385 days, time at

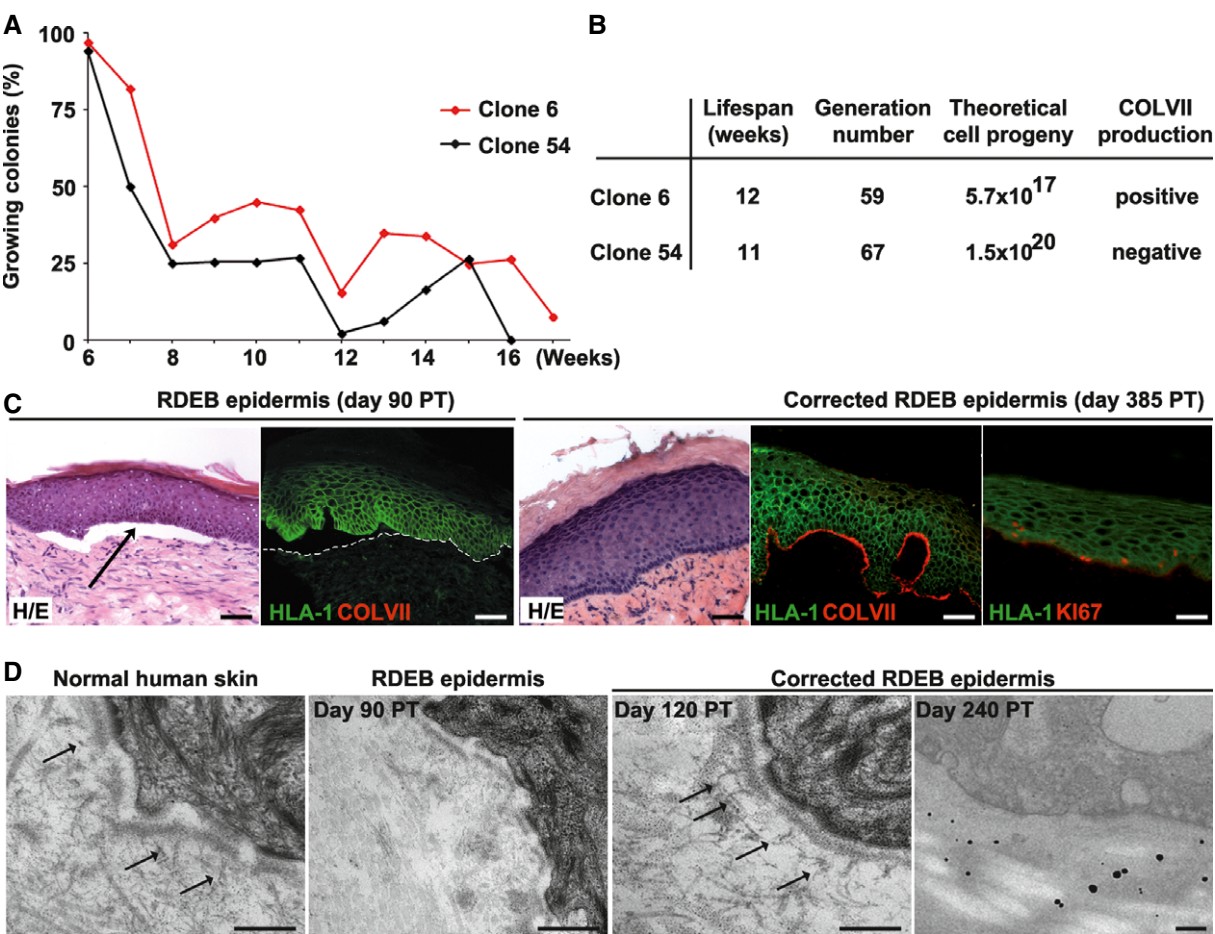

**Figure 4.  Long-term restoration of COLVII expression, generation of epidermis and anchoring fibrils by the progeny of a corrected recessive dystrophic epidermolysis bullosa (RDEB) epidermal stem cell.**

A    Serial cultivation of transduced and untransduced holoclones demonstrated that the growth potential of the stem cells was not affected by the production of COLVII. Non-COLVII-producing holoclone 54 (black lines) and COLVII-producing holoclone 6 (red lines) were serially transferred once a week until exhaustion (Rochat *et al*, 1994).

B    Theoretical number of epidermal cells available for characterisation and CEA production from corrected (clone 6) and uncorrected stem cells (clone 54) calculated from the day of cloning. The colony-forming efficiency and the percentage of growing colonies for each passage were used to calculate the population doubling, the generation number and the total progeny of isolated stem cells. Both corrected and uncorrected epidermal stem cells show high growth potential *in vitro*.

C    Immunodeficient SCID mice were transplanted with untransduced cultured RDEB keratinocytes (left) or the COLVII-secreting holoclone (clone 6) (right). Punch biopsies were obtained at various times post-transplantation (PT), stained with haematoxylin/eosine (H/E), for human leucocyte antigen-1 (HLA-1) (green) and human COLVII (red). RDEB keratinocytes generated a HLA-1-positive epidermis that adhered poorly to the dermo-epidermal junction (DEJ) (arrow indicates a blister) and absence of COLVII (dotted line delimits the dermis), whereas the corrected keratinocytes produced an epidermis that adhered to the dermis and deposited COLVII (red) at the DEJ. Note the presence of KI67-positive keratinocytes (red) in the basal and suprabasal layers in the corrected epidermis at 385 days post-transplantation, indicating that transplanted cells had self-renewed for more than a year. Scale bar: 50 μm.

D    Transmission electron microscopy (TEM) demonstrated the presence of anchoring fibrils (arrows) at the DEJ of the corrected epidermis (middle right panel) and in a normal human skin biopsy (left panel). Note the absence of anchoring fibrils in the RDEB epidermis (middle left panel). Detection of human COLVII by immunogold staining in the DEJ of corrected epidermis (right panel). Scale bar: 250 nm.

which the experiment was terminated because of the age of recipient mice. Proliferative KI67-positive keratinocytes were detected in the basal cell layer of the regenerated epidermis, indicating self-renewal and COLVII deposition were observed at the dermo-epidermal junction in biopsies taken at various time during the experiment (Fig 4C). Electron microscopy was used to determine whether the COLVII produced by the corrected RDEB keratinocytes could participate in the formation of anchoring fibrils. Numerous anchoring fibrils containing human COLVII were observed at the basement membrane by immunogold staining (Fig 4D), whereas no

anchoring fibrils were observed in RDEB transplants. Collectively, these experiments demonstrate that the progeny of a genetically corrected stem cell is capable of generating a self-renewing COLVII-producing epidermis for more than a year, demonstrating the feasibility of a single stem cell strategy for *ex vivo* gene therapy.

## Safety assessment of RDEB-corrected stem cells

On the basis of the scientific literature (Fink, 2009; Taylor *et al*, 2010; Goldring *et al*, 2011; Daley, 2012; Scadden & Srivastava, 2012)

and regulatory affair guidelines [http://www.fda.gov/downloads/Drugs/GuidanceComplianceRegulatoryInformation/Guidances/UCM070273.pdf (FDA, 2008); http://www.isscr.org/docs/guidelines/isscrglclinicaltrans.pdf (ISSCR, 2008)], we anticipated that the progeny of a genetically corrected stem cell should meet a number of safety criteria (Table 1). Accordingly, clones selected at the initial phase of the experimental protocol were put through a battery of tests. First, we determined the number and sites of proviral integrations by ligation-mediated PCR (LM-PCR) on transduced clones 6, 22 and 54. To perform LM-PCR, transduced keratinocytes were subcultured twice in the absence of feeder cells. Two integrations were found in transduced clone 6 and in clone 54 (Supplementary Table S2). Proviruses were inserted in the first intron or in the promoter region of hit genes as previously observed (Montini *et al*, 2006; Cattoglio *et al*, 2010). No clear data were obtained by LM-PCR for clone 22 due to remaining contamination with murine sequences from the feeder layer. To confirm this, we analysed clone 6 by fluorescence *in situ* hybridisation (FISH) but not clone 54 since it did not produce COLVII. FISH analysis demonstrated a specific hybridisation of *COL7A1* cDNA on five chromosomes, one signal on each chromosome 3, corresponding to endogenous *COL7A1*, one signal on a chromosome 2, one on a chromosome 22, confirming the LM-PCR results, and revealed an additional signal on a chromosome 11 (Fig 5A). We then investigated the integration sites by whole-genome sequencing. Next-generation sequencing (NGS) data analyses confirmed FISH results and permitted to estimate the location of the three proviral integrations (chromosomes 2, 22 and 11) (Fig 5B). Raw data are accessible on http://ncbi.nlm.nih.gov/sra/ accession number SRP050326. Integration sites were then individually characterised by conventional sequencing with a definition at a base-pair level (Fig 5B). Sequencing data confirmed the integrations identified by LM-PCR on chromosomes 2 and 22 and confirmed the integration on chromosome 11 discovered by FISH. Most importantly, these experiments demonstrated that the proviruses did not integrate in or in the vicinity of known oncogenes, therefore diminishing or even alleviating the risk of uncontrolled cell proliferation. One provirus was inserted in the first intron of the *DARS* gene (chromosome 2) (Fig 5B and Supplementary Table S2) that codes for a sub-unit of a metabolic enzyme (Taft *et al*, 2013). We thus determined by RT–PCR that the level of *DARS* expression was unchanged in the corrected stem cell compared to uninfected cells (Fig 5C). We also used NGS data to gain information on insertion/deletion (indel) and single nucleotide polymorphism (SNP) identified in the *DARS* sequence of clone 6. None of the annotated indel or SNP was associated with disorders described in the genomic database (GWAS catalogue and Cosmic GS5). However, two undescribed indels and one SNP were identified (Fig 5D). The SNP at position chr2:136673894 hits exon 11 with no consequence as the new codon coded for the same amino acid (synonymous SNP).

Next, we investigated the transduced stem cells for immortalisation, neoplastic transformation and dissemination. First, the level of several proteins associated with immortalisation (Hanahan & Weinberg, 2011) was determined (Fig 6A). Levels of p16, p21, p53 and RAS (Fig 6A upper panel) and the phosphorylation of pRb (Fig 6A lower panel) were similar in corrected stem cells (clone 6) and in uninfected cells (RDEB) compared to squamous cell carcinoma cell line (SCC-13). These results demonstrated that corrected stem cells did not acquire excessive proliferative potential and did not

evade growth suppression, further supporting the results of the serial transfer (Fig 4A) demonstrating that the cells had a finite lifespan. Likewise, karyotype analysis demonstrated diploidy and absence of chromosomal translocation in clone 6 and clone 22 in early and late passages (Fig 6B). Telomere measurement by quantitative PCR (Cawthon, 2002) did not show significant difference between clone 6 and RDEB keratinocytes at passage XIV (mass culture) (Supplementary Fig S7). Clones 6, 22 (COLVII positive) and 54 (COLVII negative) were then subcutaneously injected into immunodeficient athymic mice to evaluate their tumorigenic potency. None of the clones formed tumours when compared to a squamous cell carcinoma cell line (SCC-13), which formed sizeable tumours 2 months after injection (Fig 6C). To test for the disseminative potential of the genetically corrected keratinocytes, blood, gonads and other internal organs of the mice successfully transplanted with clone 6, clone 22 or non-diseased stem cells (YF29) were harvested and submitted to PCR analysis for human *COL7A1*-specific sequences. All tests were negative (Fig 6D). Collectively, these experiments demonstrate that it is possible to identify and expand transduced stem cells which progeny fulfil strict safety requirements to a level of confidence never reached before.

## Discussion

The strategy described here aims at narrowing the risk associated with *ex vivo* gene therapy as the medicinal product is thoroughly characterised before its use in the clinic. The validation process meets all safety recommendations of the International Society for Stem Cell Research (ISSCR) and the scientific community (Taylor *et al*, 2010; Goldring *et al*, 2011; Daley, 2012; Scadden & Srivastava, 2012). Hence, this strategy should help regulatory agencies in their task encouraging innovation while protecting patients (Buchholz *et al*, 2012; Abbott, 2013; Bianco *et al*, 2013a,b; Gaspar *et al*, 2013). Importantly, a successful clonal strategy necessitates the combination of efficient transduction, a superior culture system to efficiently expand the founder stem cells and a performant transplantation procedure. Our experiments make this clear demonstration using skin and validate a clonal strategy as the best option for safe *ex vivo* gene therapy by today standards.

The risk of insertional mutagenesis and oncogenesis (Moiani *et al*, 2012; Trono, 2012), whether the medicinal cells are adult stem cells (Hacein-Bey-Abina *et al*, 2002; Mavilio *et al*, 2006; Barrandon, 2007) or ES-iPS-derived cells (Hockemeyer *et al*, 2009), is a major concern in *ex vivo* gene therapy. In fact, the use of epidermal cells derived from genetically autologous-corrected iPS for treating RDEB patients has been recently proposed (Itoh *et al*, 2011) but this strategy still relies on the use of recombinant viral vectors to deliver the medicinal gene. Consequently, it does not alleviate the risk of insertional mutagenesis and the need for a clonal strategy. The fact that rearrangements of the *COL7A1* transgene can be identified in clones, and not in the mass culture from which the clones had been isolated, clearly demonstrates that a clonal strategy brings a higher degree of sensitivity and therefore safety. This said, it is worth-noting that rearrangements are not necessarily deleterious as clone 6 produced perfectly functional COLVII that, for months, participates in the formation of anchoring fibrils. The determination of the karyotype, tumorigenicity, proviral integrations and potential rearrangements before the cells are transplanted onto the patient confers additional levels of safety. Only a

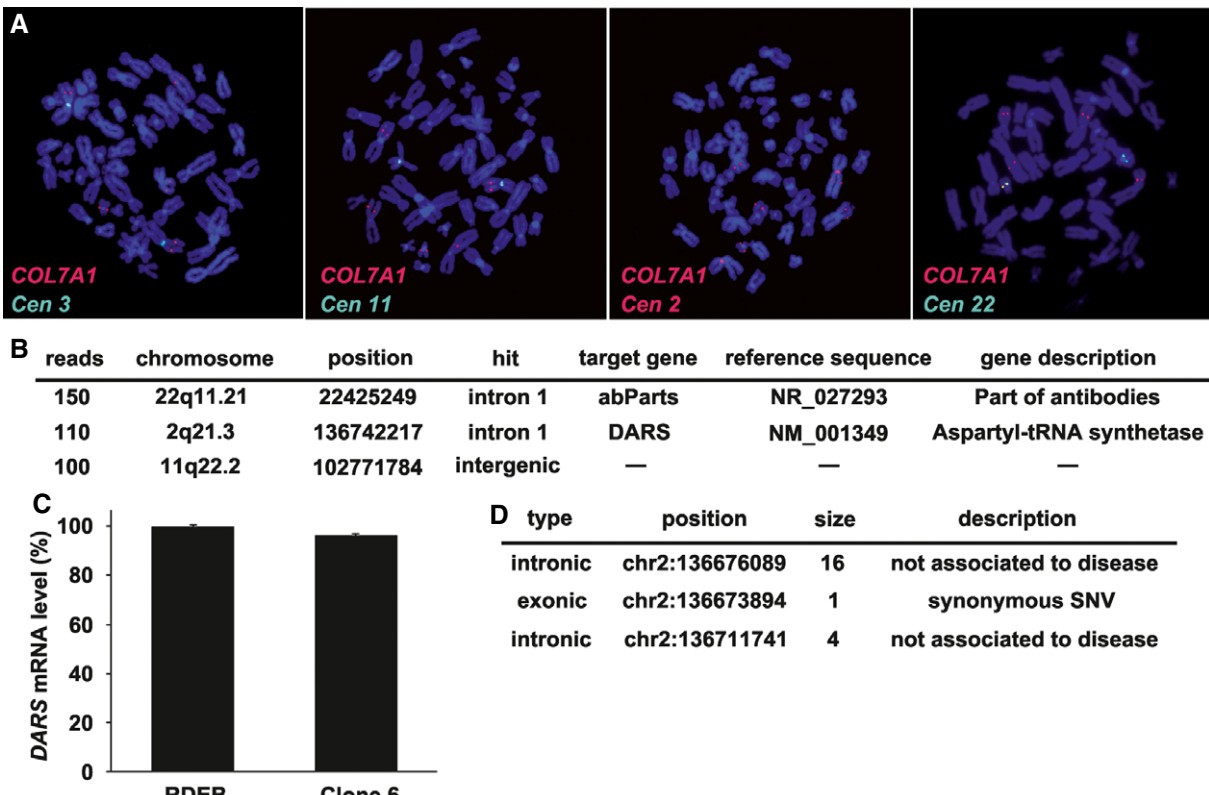

**Figure 5.   Characterisation of proviral integrations in corrected stem cells.**

A   Visualisation of *COL7A1* sequence by FISH analysis in corrected stem cells at passage XII (clone 6). Five specific signals (red) were detected on five different chromosomes. As expected, a *COL7A1* probe (red) hybridised to the two endogenous *COL7A1* alleles located on the chromosomes 3 as identified by a specific centromeric probe (Cen 3) and to three other chromosomes identified as chromosomes 2, 11 and 22 by means of specific centromeric probes (Cen 2, Cen 11 and Cen 22, respectively). The latter corresponded to proviruses.

B   Determination of proviral integration sites in corrected stem cell. Genomic DNA from clone 6 was submitted to whole-genome sequencing together with PCR and subsequent Sanger sequencing to identify the integration sites to the base-pair level. Mate pairs that span from the viral sequence to a human chromosome were extracted and used to estimate integration regions. The reads were mapped to the hg19 reference sequence. Three integration sites were uncovered: one in chromosome 2, one in chromosome 11 and one in chromosome 22, and targeted genes were identified.

C   Analysis of the level of expression of targeted genes in corrected stem cells by quantitative RT–PCR. *DARS* was not significantly changed in clone 6 compared to the cells before transduction. Primers used annealed downstream of the proviral integration site. Error bars represent the standard deviation of three replicates.

D   Identification of sequence abnormalities in the two alleles of the *DARS*-targeted gene in clone 6 by NGS. Reads were mapped to the hg19 reference sequence. Small insertion–deletions and SNP calling were performed and compared to GWAS databases. The mapping highlighted two indels and one SNP in the *DARS* genomic sequence. The indels were not associated with disease and the SNP was synonymous (CCU–CCG both codons correspond to proline).

clonal strategy permits the transplantation of a safe population of transduced cells, an otherwise impossible task with a mass culture. Combining new generations of SIN retroviral or lentiviral shuttle vectors with a clonal strategy (Almarza *et al*, 2011) is certainly the most efficient way to minimise the risk of insertional oncogenesis (Hacein-Bey-Abina *et al*, 2008; Howe *et al*, 2008). A clonal strategy can also be combined with zinc finger nucleases and integrase-defective lentiviral vectors (Lombardo *et al*, 2007; Genovese *et al*, 2014), site-specific integration and tailoring of cassette design for sustainable gene transfer (Lombardo *et al*, 2011), TALE nucleases (Hockemeyer *et al*, 2011; Reyon *et al*, 2012) or vectors with an adaptive safety that permits the selective destruction of unwanted recombinant cells (Di Stasi *et al*, 2011).

Efficacy of a medicinal stem cell product depends on permanent stem cell engraftment, the capacity of the engrafted stem cells to self-renew and to robustly produce the protein of interest. We have demonstrated that it is feasible to obtain an efficacious medicinal product from a single human epidermal stem cell, whose progeny generates a functional epidermis that (i) self-renews for more than a year when transplanted onto immunodeficient mice and (ii) robustly produces the medicinal protein which incorporates into anchoring fibrils. Consequently, the recombinant epidermis is firmly attached onto the dermis and does not blister. Nevertheless, there are important points that need to be emphasised. Maintenance of stem-ness is critical and can only be achieved by minimising cell stress during the entire *ex vivo* procedure. It is well known that cultured epidermal stem cells respond to stress by undergoing clonal conversion, that is converting a stem cell into a transient amplifying cell of restricted growth potential (Barrandon *et al*, 2012; Rochat *et al*, 2013); consequently, culture exhaustion rapidly occurs, jeopardising the production of transplantable medicinal cells and engraftment. To limit cell stress, one must first use state-of-the-art culture conditions to maintain the number of cell divisions to a minimum to reduce telomere shortening and culture aging. We calculated

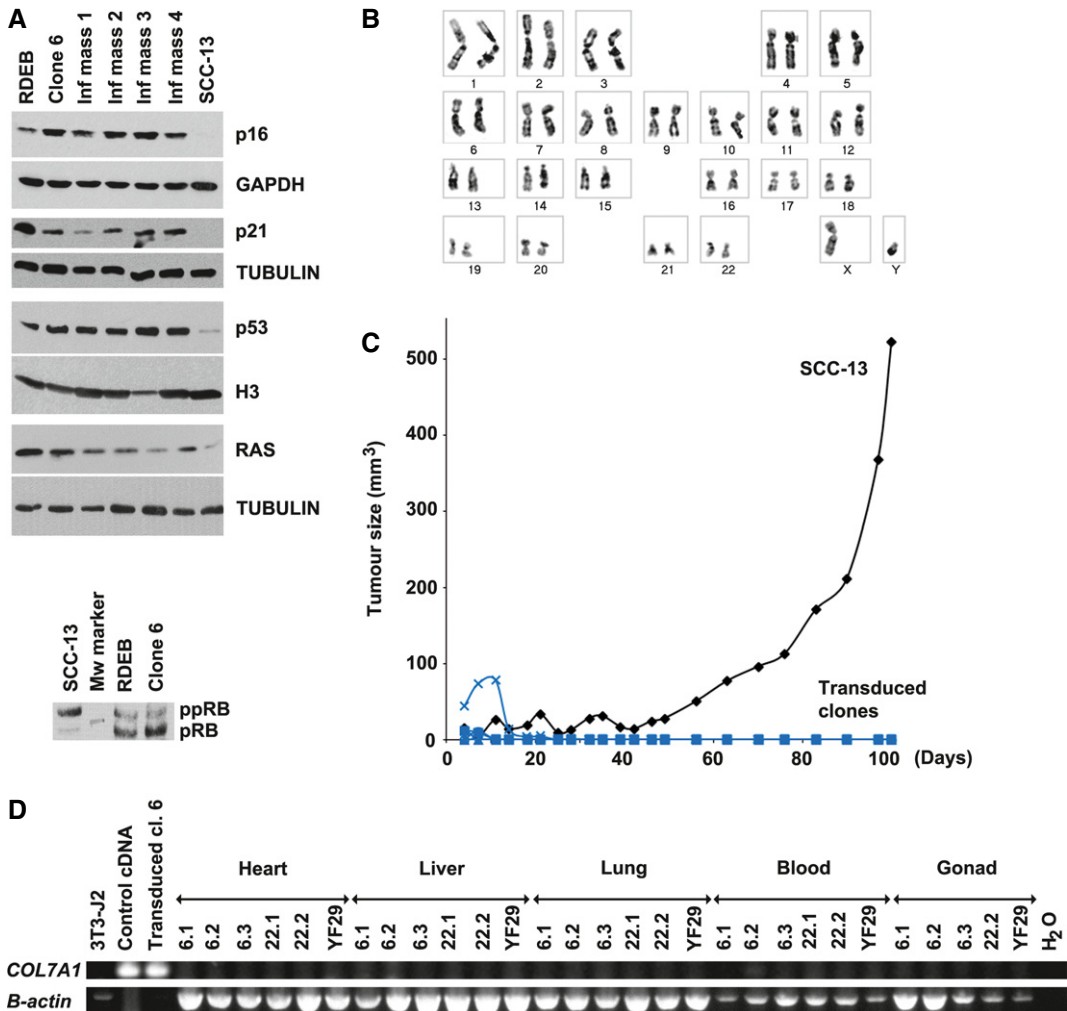

**Figure 6.  Assessment of immortalisation, tumorigenicity and disseminative potential of the corrected stem cells.**

A   Upper panel: expression of proteins associated with the acquisition of immortalisation process involved in senescence, in evasion of growth suppression and in apoptosis (Hanahan & Weinberg, 2011). The level of P53, P21, RAS and P16 was similar in clone 6, untransduced cells and mass culture from four independent infections, significantly different from the squamous cell carcinoma cell line SCC-13. Lower panel: the phosphorylation state of the PRB restriction point was maintained in clone 6 and untransduced recessive dystrophic epidermolysis bullosa (RDEB) cells, whereas PRB was heavily phosphorylated in transformed cells (SCC-13). Appropriate loading controls were used for each cellular extract (GAPDH for cytoplasmic extracts, histone H3 for nuclear extracts and tubulin for whole-cell extracts. H3 for histone H3, ppRB for hyperphosphorylated pRB and pRB for hypophosphorylated PRB).

B   Transduced clone 6 had a diploid karyotype at passage XVI (see also Fig 5A).

C   Transduced clones were not tumorigenic. The tumorigenic potential of the clones was tested by tumour formation in nude mice. Transduced RDEB holoclones (blue) [clone 6 (square, n = 4) passage X, clone 22 (circle, n = 2) passage XI, clone 54 (cross, n = 4) passage X] were not tumorigenic when injected subcutaneously in athymic mice as were untransduced RDEB keratinocytes (passage VII). SCC-13, a squamous cell carcinoma cell line (black) (lozenge, n = 2), was used as a positive control.

D   To test whether the transduced RDEB keratinocytes had disseminated after the generation of an epidermis onto SCID mice (see Fig 4C), the internal organs of the recipient mice (3 mice for clone 6 and 2 mice for clone 22) were harvested 385 days post-transplantation and analysed for COL7A1 proviral sequences. No COL7A1 sequence was detected. A mouse transplanted with a holoclone obtained from a healthy donor (YF29) was used as an internal control. PCR-positive controls were genomic DNA from transduced cells (holoclone 6) and cDNAs isolated from healthy keratinocytes. B-actin was run as a loading control.

that the entire procedure from initial isolation of a population of epidermal stem cells from the biopsy to the transplantation of cloned genetically corrected CEA necessitates a maximum of 30 divisions. This is a reasonable number when a holoclone can undergo at least 180 doublings in culture (Rochat *et al*, 1994; Mathor *et al*, 1996). Second, one must favour a friendly cell-sorting technology over fluorescence-activated cell sorting (FACS) as it is well documented that the FACS procedure is a source of stress for epidermal stem cells (Barrandon *et al*, 2012). By combining a gentle manual

cloning strategy with the selection of small cells (Barrandon & Green, 1985), we regularly achieve cloning efficiency of at least 50%, often producing more clones than one can reasonably analyse. This said, the generation of a large number of clones can undoubtedly increase the chance of obtaining one or several transduced stem cells that meet the selection criteria. For this purpose, we are developing a high-throughput system that combines the gentle isolation of numerous single cells together with an efficient identification of clones secreting the protein of interest. The capacity to prepare

transplants from a pool of fully characterised recombinant human stem cells shall also address the question of stem cell renewal and clonal dominance in human epidermis.

In conclusion, our experiments demonstrate for the first time the feasibility of a clonal strategy for *ex vivo* gene therapy. State-of-the-art transduction and culture technologies together with master and working cell banks should ensure the quality and the reproducibility of the medicinal cells (Scadden & Srivastava, 2012). Most importantly, implementing a safer procedure that reduces the odds of serious deleterious events should help the decision-making process to perform *ex vivo* gene therapy (Cavazzana-Calvo *et al*, 2004; Mavilio, 2012); this is particularly important in the case of transplanting young children, as is the goal in RDEB.

# Materials and Methods

### Cell culture

Human keratinocytes or fibroblasts were isolated from the biopsy of the wrist of a 4-year-old RDEB patient (Hilal *et al*, 1993), from the foreskin of a newborn (YF29) and from a 42-year-old female (OR-CA) as described in the Supplementary Materials and Methods. Studies were performed from frozen cells isolated from a biopsy obtained with informed consent; informed consent was obtained from patient for publication of the current study. The experiments conformed to the principles set out in the WMA Declaration of Helsinki and the Department of Health and Human Services Belmont Report. Keratinocytes and the squamous cell carcinoma cell line SCC-13 (Rheinwald & Beckett, 1981) were cultured onto a feeder layer of lethally irradiated 3T3-J2 cells (Rheinwald & Green, 1975) in medium supplemented as described (Rochat *et al*, 1994; Ronfard *et al*, 2000). RDEB fibroblasts were cultured in DMEM supplemented with 10% foetal calf serum (FCS) (Hyclone). The Flp293A-E1aColVII1 was cultured in DMEM supplemented with 10% heat-inactivated FCS. All cultures were incubated at 37°C in a 10% $CO_2$ atmosphere.

### Clonal analysis and serial transfer

Single cells were isolated as described (Barrandon & Green, 1985). Briefly, 150 individual cells trypsinised from a mass of infected cells were aspirated into a Pasteur pipette under a Zeiss Axiovert inverted microscope using a 10× objective and immediately inoculated into a 35-mm size Petri dish already containing lethally irradiated 3T3-J2 cells. Clonal types were determined as described (Barrandon & Green, 1987). CFE and serial transfer were performed as described (Rochat *et al*, 1994) in the Supplementary Materials and Methods.

### Vector production and retroviral infection

The Flp293A-E1aColVII1 producer clone was generated by Genethon, Evry, France, as described (Schucht *et al*, 2006). Infection was performed as described in the Supplementary Materials and Methods. Briefly, $2 \times 10^4$ keratinocytes from early passage were seeded onto the 3T3-J2. Infection was performed 16–24 h later with a theoretical multiplicity of infection (MOI) of 10. The infection process was repeated 24 h after the first round.

### Transplantation of cultured epithelium

A fibrin-based matrix containing RDEB fibroblasts was prepared as described (Larcher *et al*, 2007). A total of $10^5$ keratinocytes were then seeded on top of the fibrin gel, grown to confluence in culture medium supplemented with 150 IU/ml aprotinin (Trasylol, Bayer) and transplanted onto the back of 8- to 10-week-old Fox-Chase SCID mice (Charles River Laboratories) following the protocol detailed in the Supplementary Materials and Methods. Animal work was authorised by the veterinarian canton de Vaud authorization 2033. Animals were handled according to ethical standards by qualified persons. Immunodeficient mice (SCID and athymic) were housed in an official animal facility, in compliance with Swiss governmental guidelines. Studies were monitored using an online organisational tool. Grafts were harvested at different time points and processed for histology, immunocytochemistry or electron microscopy. All experiments performed conform to NIH, MRC and ARRIVAL guidelines for animal welfare.

### Immunodetection and histology

Immunostainings were performed following standard protocols (antibodies listed in the Supplementary Materials and Methods). Images of sections from skin biopsies were false-coloured in green (AF568) and red (Hoechst). Histological analyses were performed in parallel to immunostaining.

### Electron microscopy

EM was performed according to standard protocols detailed in the Supplementary Materials and Methods. Sections were examined with a Phillips CM10 transmission electron microscope at a filament voltage of 80 kV. Images were collected using a CCD camera (Morada, SIS). For immuno-electron microscopy, punches were fixed and vibratome (Leica VT100) sectioned into 50 μm slices, cryoprotected and freeze-thawed twice in liquid nitrogen. Immunodetection was performed as described in the Supplementary Materials and Methods.

### Western blotting

Immunoblots were performed on concentrated cell supernatants and cell extracts as described in the Supplementary Materials and Methods. Antibodies are listed in the Supplementary Materials and Methods.

### Karyotype and fluorescence *in situ* hybridisation

Karyotypes were made by ChromBios, Germany. FISH was performed as previously described (Pinkel *et al*, 1988) using a *COL7A1* cDNA probe labelled with Spectrum Red as described in the Supplementary Materials and Methods.

### Tumorigenic and dissemination assays

A total of $10^6$ transduced keratinocytes or SCC-13 cells were inoculated subcutaneously into the ventral flanks of 7- to 9-week-old athymic Swiss $Nu^{-/-}$ mice (Charles Rivers Laboratories) with a

21-gauge needle. Tumour formation was monitored twice a week and their diameter measured. For dissemination experiments, internal organs of immunodeficient Fox-Chase SCID mice (Charles River Laboratories) transplanted with recombinant COLVII keratinocytes were harvested at the termination of the experiment. DNA was extracted using the QIAamp DNA mini kit (Qiagen) according to the manufacturer's instructions. One hundred nanograms of DNA was submitted to PCR (BioConcept) amplification with GoTaq PCR reagent kit (Promega) with primers listed in the Supplementary Materials and Methods. PCR products were sequenced (Fasteris, Switzerland).

### Quantitative reverse-transcriptase PCR

Cells were lysed in TRIzol (Invitrogen) and RNA extracted using the RNA extraction kit (Qiagen) according to the manufacturer's instructions. Total cDNAs were obtained as previously described (Bonfanti *et al*, 2010), and quantitative PCRs were performed with Light-Cycler FastStart DNA Master SYBR Green I kit (Roche Diagnostics) in capillaries according to the manufacturer's instructions. Primers and programmes are listed in the Supplementary Materials and Methods. All PCR products were sequenced (Fasteris, Switzerland).

### Southern blotting

Genomic DNA was extracted with QiAmp DNA kit (Qiagen) according to the manufacturer's instructions. Ten micrograms of DNA was codigested with SpeI/EcoRV HF (New England Biolabs) and loaded on a 0.8% agarose (Promega) gel. After treatment described in the Supplementary Materials and Methods, DNA was transferred onto a Hybond XL (GE Healthcare Amersham) membrane by capillarity and fixed with UV (Stratalinker, Stratagene). The membrane was then hybridised with a radiolabelled probe as described in the Supplementary Materials and Methods. The probe was obtained from PCR amplification of pTOPO*COL7A1* with primers described in the Supplementary Materials and Methods.

### Whole-genome sequencing

Whole-genome sequencing was performed by Microsynth, Switzerland. Raw data are accessible on http://ncbi.nlm.nih.gov/sra/ accession number SRP050326. Briefly, 3 μg genomic DNA from clone 6 was sequenced using SOLiD 5500xl (Life Technologies) mate-pair reads. The reads were mapped to the hg19 reference sequence and to the vector reference sequence from the viral producer clone Flp293A-E1aColVII1. Mate pairs that span from the viral sequence to a human chromosome were extracted. Specific primers for the predicted integration sites were designed, and amplified products were sequenced to resolve the proviral integration sites at the base-pair level. Based on the mapping, SNP calling was performed and SNP were compared to dbSNP. Mapping, SNP and small indel calling were performed using LifeScope 2.5.1 with standard parameters. Mapping was viewed with UCSC Genome Browser.

A Bam file containing the hg19 mapping of the whole-genome sequencing of clone 6 can be downloaded from http://biorepo.

**The paper explained**

**Problem**

Decision to embark in a gene therapy trial balances risk and benefit. Despite indisputable success of several gene therapy clinical trials using stem cells genetically corrected *ex vivo*, serious complications happened with some resulting in patients' death. These complications resulted from insertional mutagenesis and clonal dominance that led to irreversible cellular transformation. Much effort has been made to design safe viral vectors and non-viral strategies. Although the most recent technologies offer powerful tools to identify thousands of proviral integration sites in engrafted cells, nobody has ever fully characterised transduced medicinal cells before transplantation.

**Results**

We have designed a strategy to assess the safety of *ex vivo* gene therapy before autologous transduced cells are transplanted onto the patient. We took advantage of our extensive experience in clonal analysis and in the cultivation of autologous epidermal cells suitable for long-term regeneration of epidermis to treat extensive third-degree burn wounds. Our single stem cell strategy provides a clonal progeny large enough to thoroughly assess stemness and safety using an array of cell and molecular assays, and to produce enough transplantable autologous epidermal stem cells to treat large areas of diseased skin. As a proof of concept, we choose recessive dystrophic epidermolysis bullosa (RDEB), a horrendous hereditary blistering skin disease linked to the absence of type VII collagen, a protein that participates in the anchoring of epidermis to dermis through the formation of anchoring fibrils. We infected RDEB keratinocytes with a self-inactivating retroviral vector bearing a *COL7A1* cDNA and cloned the infected cells using a cell-friendly method compatible with the maintenance of stem cell properties. Each clone of transduced RDEB epidermal stem cells was individually expanded and characterised on stringent criteria linked to stemness (long-term renewal, long-term regeneration of epidermis, long-term production of the medicinal protein COLVII, long-term reconstitution of anchoring fibrils) and safety (immortalisation, neoplastic transformation and dissemination, karyotype, precise determination of proviral integrations, whole-genome sequencing). Our results demonstrate that only few transduced cells meet stringent stemness and safety criteria, while the vast majority of cells do not.

**Impact**

We demonstrate that a clonal strategy brings a level of safety that cannot be obtained otherwise in *ex vivo* gene therapy. It also makes it possible to envision exciting gene-editing technologies.

epfl.ch/biorepo/public/public_link?sha1=3da4be0675e2a56b6d794 b51e82ecad821891b6a.

A bed file suitable for displaying the bam file and the insertions sites on the UCSC genome browser is available at http://biorepo. epfl.ch/biorepo/public/public_link?m_id = 6603&sha1= b67c31d63d 2f265e905f4e3b7048595f946d80e3.

### For more information

ORPHANET: http://www.orpha.net/consor/cgi-bin/index.php;
DEBRA: http://www.debra-international.org/homepage.html;
ISSCR: http://www.isscr.org/;
Fondation enfants papillons: http://www.enfants-papillons.ch/.

**Supplementary information** for this article is available online: http://embomolmed.embopress.org

## Acknowledgements

We are grateful to Genethon for providing us with retroviral supernatants through the Therapeuskin consortium, to Dagmar Wirth for the Flp293A-E1aColVII1 producer clone and to Alain Hovnanian for stimulating discussions. We are also grateful to Steeve Vermot and Jeanne Vannod for excellent technical help, to Patrick Reichenbach and Giulietta Maruggi for the help in Southern blotting, to Matthias Titeux for sharing his grafting protocol and to Stéphanie Rosset for technical assistance with TEM. We also thank Olivier Dormond for pRb, p21, Ras and GAPDH antibodies, Alessandro Amici for help with the tumorigenic assay and critical reading of the manuscript, Jean-Daniel Tissot for supplying us with human plasma, Alvaro Baptista and the Department of Pathology at the CHUV with skin samples and Rainer Follador from Microsynth for helpful discussion. We are particularly grateful to continuing support from La Fondation Enfants Papillons and Dr Elisabeth Gianadda, DEBRA-CH and the patient's family. YB was supported by the EPFL, the CHUV and the European Commission through the 6th (Therapeuskin) and 7[th] (Opti-Stem) framework programmes. FM was supported by the Italian Ministry of Health, the Progetto Malattie Rare (RF-EMR-2008-1210900) and the European Research Council (ERC-2010-AdG, GT-Skin).

## Author contributions

SD-GL performed the experiments, organised the collaborative work and interpreted the data. ARo initiated the project, performed the cloning experiments and interpreted the data, NG helped with transplantation experiments and ES-D with regulatory aspects. GK performed and interpreted EM experiments, and ARe and FM performed and interpreted LM-PCR. DM, NBS and JSB performed and interpreted the genetic analyses. SB and JR analysed and interpreted the bioinformatic data. AZ performed and analysed the telomere length assay. YB supervised the project, and SD-GL and YB wrote the paper.

## Conflict of interest

The authors declare that they have no conflict of interest.

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
