## [Review Process File · EMBO Molecular Medicine]

A Single Epidermal Stem Cell Strategy for Safe Ex Vivo Gene Therapy

Stéphanie Droz-Georget Lathion, Ariane Rochat, Graham Knott, Alessandra Recchia, Danielle Martinet, Sara Benmohammed, Nicolas Grasset, Andrea Zaffalon, Nathalie Besuchet Schmutz, Emmanuelle Savioz-Dayer, Jacques S. Beckmann, Jacques Rougemont, Fulvio Mavilio and Yann Barrandon

Corresponding author: Yann Barrandon, Ecole Polytechnique Fédérale de Lausanne

Review timeline:	Submission date:	18 June 2014
	Editorial Decision:	17 January 2015
	Revision received:	23 July 2014
	Editorial Decision:	20 November 2014
	Revision received:	08 January 2015
	Accepted:	21 January 2015

Transaction Report:

Editor: Céline Carret

1st Editorial Decision

17 January 2015

Thank you for the submission of your manuscript to EMBO Molecular Medicine. We have now heard back from the three referees whom we asked to evaluate your manuscript. Although the referees find the study of interest, they also raise a number of concerns that should be addressed in the next version of your article.

As you will see, all three referees praise the high-quality and importance of the findings for stem cells based-regenerative medicine. While referee 1 only has a few minor concerns, referee 3 requests further explanations and suggest depositing your WGS results on a publicly available repository, which we would also insist upon as this is mandatory for publication. We would like you to focus on addressing the concerns of referees 1 and 3. In addition, if you have data to address the concerns raised by referee 2, we would encourage you to add them in, but for a proof-of-principle type of article, human transplants are not strictly necessary.

Should you be able to address these comments, we would be happy to consider a revised manuscript.

Please note that it is EMBO Molecular Medicine policy to allow a single round of revision in order to avoid the delayed publication of research findings. Consequently, acceptance or rejection of the manuscript will depend on the completeness of your responses included in the next version of the manuscript.

I look forward to receiving your revised manuscript.

***** Reviewer's comments *****

Referee #1 (Remarks):

While several studies have reported successful ex vivo gene therapy for epidermolysis bullosa (EBs), the approach used were not performed on cells originating from a single cells, therefore precluding the extensive characterization of the cells grafted, which can results in the presence of a fraction of cells that are unsafe for cell therapy. In this regard, using a clones generated from a single cell allow extensive characterization of the clones before transplantation, which is critical for clinical safety.

Recessive dystrophic epidermolysis bullosa (RDEBs) is a life-threatening genetic disease that is due to mutation in collagen VII and results in skin blistering disorders. Although several attempts to treat the disease have been made, they all lack either efficiency or safety which results in the death of several patients, or requires continuous immunosuppression.

In this study, Droz et al. use single cell isolation together with gene correction and extensive characterization of the generated clones to overcome these limitations and meet a state of the art level of safety. As a proof of concept, they use this approach to correct keratinocytes from RDEB patients and show that they are can produce collagen VII in vivo and show no signs of EBs.

First, Droz and colleagues designed a strategy to isolate single cells from RDEB patients and grow them in sufficient amount to allow downstream manipulation. They then characterize extensively the clones and assess their medical potential in vivo.

Droz et al. first show that both wild type and RDEB keratinocytes present similar growth characteristic in vitro and are able to form holoclones. They then show that they can efficiently correct the collagen VII gene in RDEB keratinocytes correct using a self-inactivating retrovirus (SIN). This results in the production of collagen VII at the mRNA and protein level with variable efficiency between the different clones. Using Southern Blot, they show that there is clonal heterogeneity regarding the presence of transgene and provirus recombination.

Moreover, they show that the production of collagen VII does not provide a growth advantage. Droz and colleagues show that the corrected RDEB keratinocytes are able to produce functional collagen VII and do not form epidermolysis bullosa as compared to the control.

Using FISH, whole genome sequencing as well as qPCR, they show that the transgene integrated without disturbing endogenous genes function. Droz et al. further show that these manipulations do not create genomic abnormalities, do not immortalize the cells and do no render them tumorigenic.

Overall, the study is interesting, well controlled and data of high quality. In addition, the approach used here represents a significant advance for the development of safe ex vivo gene therapy in the treatment of genetic skin diseases. We think that the manuscript fully meet the standards of EMBO Molecular Medicine with the exception of a few minor points that should be addressed before publication. These points are described down below :

1) In Figure 3 C, it should be shown more clearly which fragments correspond to the rearranged provirus.

2) In Figure 4C, the author show Ki67 in corrected RDEB epidermis but not in uncorrected RDEB epidermis. This should be presented.

3) In Figure 4D, the RDEB epidermis and corrected RDEB are compared but their time points do not match. This should be corrected. Moreover, the scale and contrast of the image at Day 240PT is different from the other pictures. This impede good comparison with the other images and should be corrected.

Referee #2 (Remarks):

This manuscript from the Barrandon lab deals with the question of feasibility of gene targeting approaches for a rare genetic skin disease, by using single clone expansion and characterization prior to transplantation. The quality of the data is very high, and the experiments are well controlled and well executed. The figures are very clear and the text is well written and easy to follow. The work demonstrated that human keratinocytes transduced with a full length collagen gene can be clonally expanded in order to produce enough cells that can be characterized for safety and efficiency, and still have enough left to rebuild skin on an immunocompromised mouse. Multiple clones can also be selected, characterized and used for transplants, thus enriching the pool of stem cells in the transplants.

In my opinion, while this is a valuable study, the work is incremental and does not raise to the substantial quality I perceive required for an EMBO journal. The major caveats are: (1) only one human cell line was tested and (2) transplants on human subject are necessary to make it a substantial advancement. While I realize that this is a major endeavor, I believe the current results should be published elsewhere in a more specialized journal.

Referee #3 (Remarks):

In this exciting manuscript, Droz-Georget Lathion and colleagues reported a proof of principle case where they isolated and characterized genetically corrected human epidermal stem cell clones and successfully applied these cells to form normal human epidermis robustly in SCID mice. This report addressed an important issue in stem cell biology, that is whether a single stem cell can be utilized for regenerative medicine. In addition, their technical approach has also provided a basis to evaluate genetically corrected stem cell prior to the application to either animal models or human patients. I only have a few minor concerns that should be addressed by the authors.

1. It has been previously demonstrated that cultured human embryonic stem cells could be contaminated by proteins derived from animal serum (Martin et al., Nature Medicine 2005). In addition, although the feeder cells (derived from mouse) were lethally irradiated, there is still a possibility that they may fuse with human epidermal stem cells. The authors should address these issues, at least, in the discussion.

2. Because there are still a considerable number of cell divisions (~30) between the isolation of epidermal stem cells to the generation of genetically corrected CEA, has the author measured the length of telomere in the cloned cells immediately prior to the grafting experiments, and how does the length compare to the normal human epidermal stem cells?

3. In Fig. 3, why did the authors measure different groups of clones in A/B vs C? Did they imply the lack of intact provirus in 3, 24 led to the lack of ColVII expression? Samples 70, 73, 75, 76 in 3C were irrelevant. Also in 3A, clones 6, 22 and 54 appear to be mosaic for ColVII expression. Did the results suggest the heterogeneity in the cloned population caused by random recombination of the inserted provirus? The authors need to address this issue.

4. The estimation for the number of cell progenies in Fig. 4B was not very accurate. First, 2^{59} and 2^{67} equal to 5.7×10^{17} and 1.47×10^{20} , respectively. Second, the authors already know the percentage of growing colonies reduced about 80% from week 6 to week 12. The authors should use a more accurate estimation to reflect how many cells will be available for epidermal grafting by the end of their experiments.

5. The whole-genome sequencing approach is excellent to identify any potential genetic alterations. However, the mapping strategy is convoluted. And the results were quite superficially presented. For example, it was clear that clone #6 has at least two versions of recombinated ColVII based on Fig. 3C. However, they were not identified in the WGS results. The authors should deposit the raw sequencing results into public database e.g. NCBI GEO. So other groups may use the data to perform more rigorous analysis e.g. to estimate the impact of retroviral infection to human genome.

1st Revision - authors' response

23 July 2014

Referee #1 (Remarks):

While several studies have reported successful ex vivo gene therapy for epidermolysis bullosa (EBs), the approach used were not performed on cells originating from a single cells, therefore precluding the extensive characterization of the cells grafted, which can results in the presence of a fraction of cells that are unsafe for cell therapy. In this regard, using a clones generated from a single cell allow extensive characterization of the clones before transplantation, which is critical for clinical safety.

Recessive dystrophic epidermolysis bullosa (RDEBs) is a life-threatening genetic disease that is due to mutation in collagen VII and results in skin blistering disorders. Although several attempts to treat the disease have been made, they all lack either efficiency or safety which results in the death of several patients, or requires continuous immunosuppression.

In this study, Droz et al. use single cell isolation together with gene correction and extensive characterization of the generated clones to overcome these limitations and meet a state of the art level of safety. As a proof of concept, they use this approach to correct keratinocytes from RDEB patients and show that they are can produce collagen VII in vivo and show no signs of EBs.

First, Droz and colleagues designed a strategy to isolate single cells from RDEB patients and grow them in sufficient amount to allow downstream manipulation. They then characterize extensively the clones and assess their medical potential in vivo.

Droz et al. first show that both wild type and RDEB keratinocytes present similar growth characteristic in vitro and are able to form holoclones. They then show that they can efficiently correct the collagen VII gene in RDEB keratinocytes correct using a self-inactivating retrovirus (SIN). This results in the production of collagen VII at the mRNA and protein level with variable efficiency between the different clones. Using Southern Blot, they show that there is clonal heterogeneity regarding the presence of transgene and provirus recombination.

Moreover, they show that the production of collagen VII does not provide a growth advantage. Droz and colleagues show that the corrected RDEB keratinocytes are able to produce functional collagen VII and do not form epidermolysis bullosa as compared to the control.

Using FISH, whole genome sequencing as well as qPCR, they show that the transgene integrated without disturbing endogenous genes function. Droz et al. further show that these manipulations do not create genomic abnormalities, do not immortalize the cells and do no render them tumorigenic.

Overall, the study is interesting, well controlled and data of high quality. In addition, the approach used here represents a significant advance for the development of safe ex vivo gene therapy in the treatment of genetic skin diseases. We think that the manuscript fully meet the standards of EMBO Molecular Medicine with the exception of a few minor points that should be addressed before publication. These points are described down below:

1) In Figure 3 C, it should be shown more clearly which fragments correspond to the rearranged provirus.

We have modified Figure 3C and identified the rearranged forms with asterisks.

2) In Figure 4C, the author show Ki67 in corrected RDEB epidermis but not in uncorrected RDEB epidermis. This should be presented.

It is difficult to obtain long-term engraftment of cultured diseased RDEB keratinocytes because of the extreme fragility of the regenerated epidermis that poorly adheres onto the grafting bed. We went back to frozen samples but failed to obtain a conclusive KI67 immunostaining because of samples degradation. We have decided not to perform again the experiment because uncorrected RDEB cells fail to engraft most of the time. Nevertheless the fact that an epidermis is present 90 days after transplantation as shown in Figure 4C clearly indicates that the epidermis regenerated from transplanted cultured RDEB has self-renewed.

3) In Figure 4D, the RDEB epidermis and corrected RDEB are compared but their time points do not match. This should be corrected. Moreover, the scale and contrast of the image at Day 240PT is different from the other pictures. This impedes good comparison with the other images and should be corrected.

The lifespan of the recipient SCID mice and the macroscopic appearance of the grafts determined the duration of an experiment. In a practical way, an experiment was terminated when recipient mice were showing signs of failure. Moreover, engraftment of corrected RDEB stem cells was much more efficient than uncorrected RDEB as mentioned above. Furthermore we wanted to keep the mice transplanted with corrected RDEB stem cells as long as possible to shown long term regeneration of epidermis and long term presence of anchoring fibrils. This is why time points of graft harvest are different and impossible to match.

The contrast of the immunogold photograph (last photograph in Figure 4D) is different from conventional TEM because the samples are treated differently (see methods). The magnification had to be adjusted to localise the particles of gold, not to examine the cell morphology.

Referee #2 (Remarks):

This manuscript from the Barrandon lab deals with the question of feasibility of gene targeting approaches for a rare genetic skin disease, by using single clone expansion and characterization prior to transplantation. The quality of the data is very high, and the experiments are well controlled and well executed. The figures are very clear and the text is well written and easy to follow. The work demonstrated that human keratinocytes transduced with a full length collagen gene can be clonally expanded in order to produce enough cells that can be characterized for safety and efficiency, and still have enough left to rebuild skin on an immunocompromised mouse. Multiple clones can also be selected, characterized and used for transplants, thus enriching the pool of stem cells in the transplants.

In my opinion, while this is a valuable study, the work is incremental and does not raise to the substantial quality I perceive required for an EMBO journal. The major caveats are: (1) only one human cell line was tested and (2) transplants on human subject are necessary to make it a substantial advancement. While I realize that this is a major endeavor, I believe the current results should be published elsewhere in a more specialized journal.

In this work, we have used several individual clones; each of them can be considered as an individual strain with its own genetic and growth capabilities. We fully agree with the reviewer that a human clinical trial is the ultimate goal of a gene therapy project. However, we believe that the clinical transplantation of corrected RDEB stem cells is out of the scope of this paper as it necessitates substantial time to get permission to experiment on human and significant time of follow-up. Furthermore, the publication of proof of principle experiments is often required by regulatory affairs as it strengthens an application for a phase I clinical trial.

Referee #3 (Remarks):

In this exciting manuscript, Droz-Georget Lathion and colleagues reported a proof of principle case where they isolated and characterized genetically corrected human epidermal stem cell clones and successfully applied these cells to form normal human epidermis robustly in SCID mice. This report addressed an important issue in stem cell biology, that is whether a single stem cell can be utilized for regenerative medicine. In addition, their technical approach has also provided a basis to evaluate genetically corrected stem cell prior to the application to either animal models or human patients. I only have a few minor concerns that should be addressed by the authors.

1. It has been previously demonstrated that cultured human embryonic stem cells could be contaminated by proteins derived from animal serum (Martin et al., Nature Medicine 2005). In addition, although the feeder cells (derived from mouse) were lethally irradiated, there is still a possibility that they may fuse with human epidermal stem cells. The authors should address these issues, at least, in the discussion.

We agree with the reviewer that the transplantation of human cells cultured in presence of fetal calf serum and an irradiated mouse feeder is a *stricto sensu* a xenotransplantation. However, the culture system that we used here is agreed upon by regulatory agencies (USA, Europe, Singapore, Australia, Japan Korea, India) for the treatment of extensive burn wounds with cultured autologous epidermal stem cells. To our knowledge, no detrimental adverse reaction has been reported. Many laboratories (including us) have tried to replace the serum and feeder cells. These experiments have been unsuccessful as human keratinocytes rapidly lose growth potential making it impossible to obtain massive expansion and *a fortiori* clonal growth.

We have not observed fusion between human keratinocytes and irradiated mouse cells as confirmed by karyotyping, *in vivo* transplantation, LM-PCR, FISH and NGS.

2. Because there are still a considerable number of cell divisions (~30) between the isolation of epidermal stem cells to the generation of genetically corrected CEA, has the author measured the length of telomere in the cloned cells immediately prior to the grafting experiments, and how does the length compare to the normal human epidermal stem cells?

We have addressed the problem of telomere length by performing q-PCR experiments according to Cawthon's protocol (Nucleic Acids Res 2002) and the results are discussed in the text and shown in Figure S7. In summary, there is no significant difference between corrected clone 6 and RDEB uncorrected keratinocytes at passage XIV as for control keratinocytes.

3. In Fig. 3, why did the authors measure different groups of clones in A/B vs C? Did they imply the lack of intact provirus in 3, 24 led to the lack of ColVII expression? Samples 70, 73, 75, 76 in 3C were irrelevant. Also in 3A, clones 6, 22 and 54 appear to be mosaic for ColVII expression. Did the results suggest the heterogeneity in the cloned population caused by random recombination of the inserted provirus? The authors need to address this issue.

We have focused on good growing clones (holoclones) as well as clones that do not express type VII collagen. We agree with the reviewer's interpretation that latter clones do not have intact provirus. As suggested, we have also removed samples 70, 73, 75, 76 in Figure 3C.

It is true that the progeny of a single cell seems to express different levels of type VII collagen as reflected by the immunofluorescence data presented in Figure 3A. Our interpretation is that it results either from a heterogeneity in the growth of the clonal population (see Barrandon and Green PNAS 1987) or/and from cells at different stages of differentiation as shown in Figure S1 B right panel (colony of non diseased keratinocyte strain (OR-CA)).

4. The estimation for the number of cell progenies in Fig. 4B was not very accurate. First, 2^{59} and 2^{67} equal to 5.7×10^{17} and 1.47×10^{20} , respectively. Second, the authors already know the percentage of growing colonies reduced about 80% from week 6 to week 12. The authors should use a more accurate estimation to reflect how many cells will be available for epidermal grafting by the end of their experiments.

We have modified the numbers accordingly in the text and in Figure 4B

5. The whole-genome sequencing approach is excellent to identify any potential genetic alterations. However, the mapping strategy is convoluted. And the results were quite superficially presented. For example, it was clear that clone #6 has at least two versions of recombinated ColVII based on Fig. 3C. However, they were not identified in the WGS results. The authors should deposit the raw sequencing results into public database e.g. NCBI GEO. So other groups may use the data to perform more rigorous analysis e.g. to estimate the impact of retroviral infection to human genome.

The reviewer is absolutely correct: clone 6 has at least two versions of recombinated ColVII based on Southern blot presented in Figure 3C. However the rearrangements could not be precisely addressed by WGS because reads cannot be attributed to a specific insertion as they are all mixed in the genomic sample. Furthermore, COL7A1 cDNA has many highly repetitive sequences, which renders the rearrangement difficult to identify.

All raw data have been made accessible. Internet links are given in Material and Methods.

2nd Editorial Decision

20 November 2014

Thank you for the submission of your revised manuscript to EMBO Molecular Medicine. We have now received the enclosed reports from the referees that were asked to re-assess it. As you will see the reviewers are now fully supportive and I am very happy to inform you that we will be able to accept your manuscript pending editorial final amendments.

Please submit your revised manuscript within two weeks. I look forward to seeing a revised form of your manuscript as soon as possible.

***** Reviewer's comments *****

Referee #1 (Remarks):

The authors addressed our initial comments and improved the quality of their manuscript through their revision. This paper will be very important for the stem cell community.

Referee #3 (Remarks):

The authors have addressed all of my concerns. I recommend acceptance of this manuscript for publication.